# All-Cellulose Composites Properties from Pre- and Post-Consumer Denim Wastes: Comparative Study

**Behnaz Baghaei** [1,*] , **Belinda Johansson** [1] , **Mikael Skrifvars** [2] **and Nawar Kadi** [1]

1 Department of Textile Technology, Faculty of Textiles, Engineering and Business, University of Borås, SE-501 90 Borås, Sweden; s183641@student.hb.se (B.J.); nawar.kadi@hb.se (N.K.)
2 Department of Resource Recovery and Building Technology, Faculty of Textiles, Engineering and Business, University of Borås, SE-501 90 Borås, Sweden; mikael.skrifvars@hb.se
* Correspondence: behnaz.baghaei@hb.se

**Abstract:** This study reports the recycling of discarded denim textiles by the production of all-cellulose composites (ACCs). Discarded denim fabrics were shredded into fibers and then made into nonwoven fabrics by carding and needle punching. The produced nonwoven fabrics were converted to ACCs by one-step and two-step methods using an ionic liquid (IL), 1-butyl-3-methyl imidazolium acetate ([BMIM][Ac]). In this study, the effect of different ACC manufacturing methods, denim fabrics with different contents (a 100% cotton denim (CO) and a blend material (cotton, poly-ester and elastane (BCO)) and reusing of IL as a recycled cellulose solvent on the mechanical pro-perties of the formed ACCs were investigated. The ACCs were characterized according to their tensile and impact properties, as well as their void content. Microscopic analysis was carried out to study the morphology of a cross-section of the formed composites. The choice of the one-step method with recycled IL, pure IL or with a blend material (BCO) had no influence on the tensile properties. Instead, the result showed that the two-step method, with and without DMSO, will influence the E-modulus but not the tensile strength. Regarding the impact properties of the samples, the only factor likely to influence the impact energy was the one-step method with CO and BCO.

**Keywords:** all-cellulose composites; end-of-life textiles; denim fabrics; ionic liquid; mechanical properties; sustainability

## 1. Introduction

Today's consumption of textiles generates a large volume of discarded textiles. It is therefore necessary to take action against and responsibility for textile waste [1]. One of the UN's sustainable development goals is "ensure sustainable consumption and production patterns" [2]. This sustainability goal does not cohere with the increasing consumption of textile products and the volume of discarded textiles ending up in landfills or incineration plants [3,4]. The development of environmentally friendly and sustainable secondary recycled materials from the discarded textiles is therefore of importance, where development is moving towards materials from renewable resources and recycled fibers, polymers and textiles [5,6].

One sector of the textile industry which has a negative impact on the environment is the life cycle of denim textiles [7]. Denim is a part of today's fast fashion and post-consumer denim waste follows this trend. For instance, a pair of Levi's 501 jeans requires 3781 L of water over its lifetime [8]. Luiken and Bouwhuis (2015) [9] claimed that the cutting waste from denim jeans production is between 10 and 15% of the used denim fabric. Furthermore, the researchers estimated that the total textile consumption of jeans is 2.16 million tons per year, while in Western Europe the collected denim textile waste accounts for 35–50% of textile waste. Most denim fabric is a woven twill fabric, consisting of 100% cotton, 60% cotton/40% polyester, 50% cotton/50% polyester,

60% polyester/40% cotton and cotton/nylon/polyester. The twill structure has better wettability and drape when compared to a plain weave [10]. Repurposing discarded textiles could potentially reduce the production of virgin textile fibers and hence reduce its environmental impact [5,11]. The repurposing does not have to be a similar textile fiber; instead, the discarded textile can be used to produce other products and materials [5]. Previous research has been done in which discarded denim was used in the development of biocomposites for structural applications [12]. Nonetheless, no research has been done where pre- and post-consumer denim textiles have been used as both a matrix and reinforcement in order to obtain a composite. To proceed with the development of composites from discarded textiles, one option could be all-cellulose composites (ACCs). ACCs were first reported by Nishino, Matsuda and Hirao in 2004 [13]. Two methods are commonly used for the production of ACCs, either a method based on dissolving the cellulosic material in a solvent which is then combined with a cellulosic reinforcement (complete dissolution/two-step method), or a method based on partially dissolving a cellulose fabric with a solvent (one-step method) [13]. Figure 1 illustrates the two different methods.

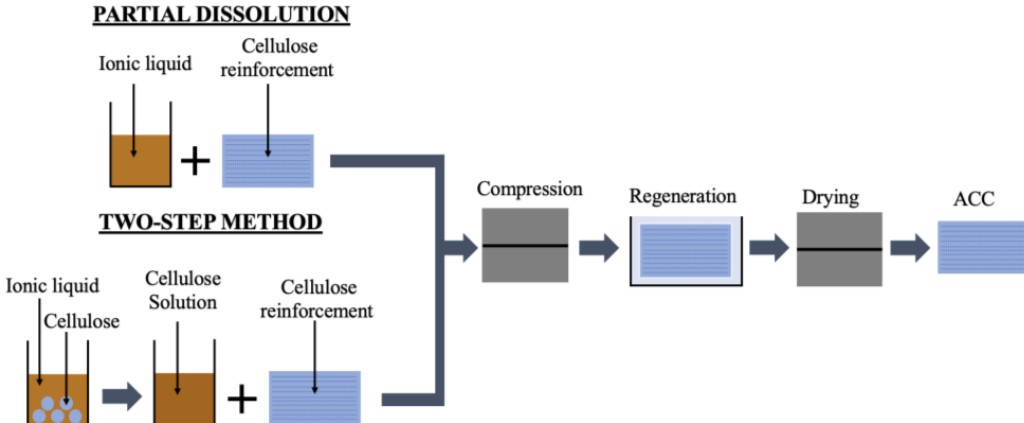

**Figure 1.** Illustration of the partial dissolution (one-step) and complete dissolution (two-step) methods.

These techniques for ACC production could be used to enable material recycling of discarded textiles into, e.g., composites for structural applications. This would increase the length of the cellulose's life, as composites are typically used in durable and long-life length applications.

The aim of this study is to create ACCs from pre- and post-consumer denim textile wastes, in the form of nonwoven fabric together with an ionic liquid to investigate if the method can be used to recycle textiles into composites.

## 2. Materials and Methods

### 2.1. Materials

In this study, two different discarded denim fabrics were used—a 100% cotton denim (CO), and a blend material (cotton, polyester and elastane) mainly composed of cotton (BCO). The pre- and post-consumer denim fabrics were provided by Texaid (Switzerland). The materials, fabric types and their area weight are given in Table 1. The ionic liquid, 1-butyl-3-methylimidazolium acetate ([BMIM][AC]) with a purity of 96%, was obtained from Sigma Aldrich. [BMIM][Ac] was used to dissolve cellulosic material in this research. Dimethyl sulfoxide (DMSO) provided by Sigma Aldrich was used as a co-solvent in the two-step method in order to decrease the viscosity of IL [14]. However, when adding a co-solvent, it can make it more difficult to recover the [BMIM][Ac] after the formation of the composite. Distilled water was used for the regeneration of the cellulose.

**Table 1.** Composition of the produced composites.

| Sample | Method | Solvent/Solution | Fiber Content (wt%) |
|:------:|:------:|:----------------:|:-------------------:|
| **CO1** | One-step | [BMIM][Ac] | - |
| **CO2** | Two-step | [BMIM][Ac] + CO | 94 |
| **CO3** | Two-step | [BMIM][Ac] + CO + DMSO | 87 |
| **BCO1** | One-step | [BMIM][Ac] | - |
| **BCO2** | Two-step | [BMIM][Ac] + CO | 90 |
| **BCO3** | Two-step | [BMIM][Ac] + CO + DMSO | 87 |
| **RILCO** | One-step | Recycled [BMIM][Ac] | - |

*2.2. Methods*

2.2.1. Sample Preparation

Mechanical shredding was chosen as the method to convert the discarded denim fabric into fibers again, since shredding enables the possibility of recycling different types of fabrics, e.g., damaged, post and pre-costumer denims. The used discarded denim was sorted according to the cellulose content into two categories: denim with 100% cotton and denim with blend materials. Zippers, buttons, seams and hard parts were removed manually with scissors from the jeans before shredding. The textiles were shredded in an NSX-FS1040 shredder, equipped with one drum with 8 mm long saw teeth, which was connected to a second shredder, an NSX-QT310, which had drums with 4 mm long saw teeth. Both machines were from Qing Dao New Shunxing Environmental Protection and Technology Co. (Qingdao, China). The fiber length measurement was done by image analysis of pictures obtained by an optical microscope (Nikon SMZ800 microscope, Tokyo, Japan). The average length of the shredded fibers before carding was 7.02 mm and after carding was 11.15 mm. The higher average length could be due to the fact that the medium carding operation eliminates the long staple fibers from the short fibers. The short fibers were collected on carding drums as can be seen in Figure 2.

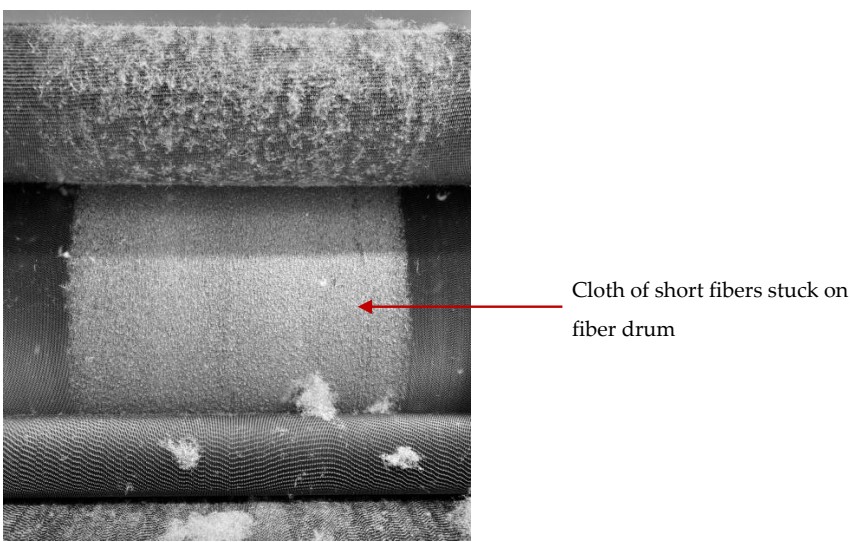

Cloth of short fibers stuck on fiber drum

**Figure 2.** Recycled denim fibers stuck between the needles on the drum in the carding machine.

Carding was performed to disentangle the fibers, position the fibers parallel to each other and produce fiber webs. In total, 60 g of fibers were collected to create each web. A Mesdan Lab Carding Machine (Brescia, Italy) was used to produce the fiber webs. The carding was done only once due to difficulties during carding where the short fibers got stuck on the fiber drum. The machine was cleaned between each carding run to prevent cloth formation on the drum, as seen in Figure 2.

To produce the nonwoven fabric, a needle punching machine with a penetration density of 0.8 needles/cm$^2$ and a stitch density of 78 stitches/cm$^2$ was used. The webs were placed in the same fiber direction as in the carding procedure.

2.2.2. Composite Laminate Manufacturing

In this study, both methods of ACCs production were used to compare if there is a difference between these methods. Table 1 shows the compositions for the produced ACCs composites. The fiber content of ACCs produced with the one-step method cannot be calculated due to the difficulty in visually distinguishing the reinforcing and matrix cellulose content in the composites.

The direction of the fibers in the nonwoven sheets was kept the same as in the carding and needle punching, which gave unidirectional composites. The total number of layers was dependent on the total weight of 50 g of material. The prepared samples were dried in an oven at 80 °C for 24 h before use to remove moisture content. A 20-ton manual bench press (Rondol Technology Ltd., Stoke On Trent, UK) was used to form the composites by compression molding. The nonwoven sheets were placed between two Teflon sheets to facilitate removal of the composite after pressing and then placed between two metal plates with a metal mold in between with a cut-out measuring 0.19 × 0.19 m$^2$.

Partial Dissolution/One-Step Method

Table 2 shows the temperature, pressure and pressing time applied for each laminate produced by the one-step and two-step methods.

**Table 2.** Processing parameters for different laminate samples produced by one-step and two-step methods.

|  | Sample | Temperature (°C) | Pressure (Pa) | Time (min) |
|---|---|---|---|---|
| **One-step method** | CO1 | 110 | 0.25, 0.50 | 60 |
|  | BCO1 | 110 | 0.25 | 60 |
|  | RILCO | 110 | 0.25 | 60 |
| **Two-step method** | CO2 | 110 | 0.125–0.375 | 60 |
|  | CO3 | 110 | 0.050–0.125 | 30 |
|  | BCO2 | 110 | 0.050–0.125 | 30, 50 |
|  | BCO3 | 110 | 0.050–0.125 | 30 |

The impregnation was performed by pouring 125 ± 1 mL [BMIM][AC] on 50 g cellulose nonwoven sheets until all nonwoven sheets were impregnated (Figure 3). The same procedure was done with the recycled [BMIM][Ac].

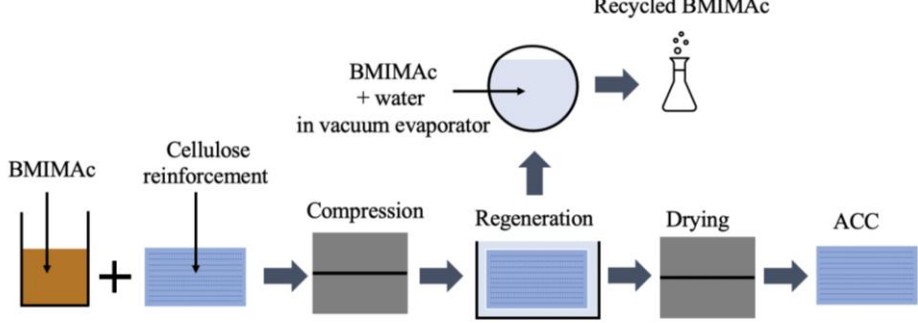

**Figure 3.** Illustration of the method for the one-step method.

Two-Step Method

The fabrics and chemicals used in the two-step method are given in Table 1. The volume of the [BMIM][Ac] was 290 ± 1 mL. The weight of cellulose for dissolution was

5.1 ± 0.1 g (1.7 wt% cellulose solution). The cellulose was dissolved in the [BMIM][Ac] by magnet rotation in a beaker at a temperature of 100 °C for 60–90 min until complete dissolution of the cellulose occurred. The solution was then spread over the nonwoven fabrics (cellulose reinforcement) with the same procedure as for the one-step method (Figure 4). The temperature in the press was 110 °C, the pressure was between 0.05 and 0.125 MPa and the processing time in the press was between 30 and 50 min.

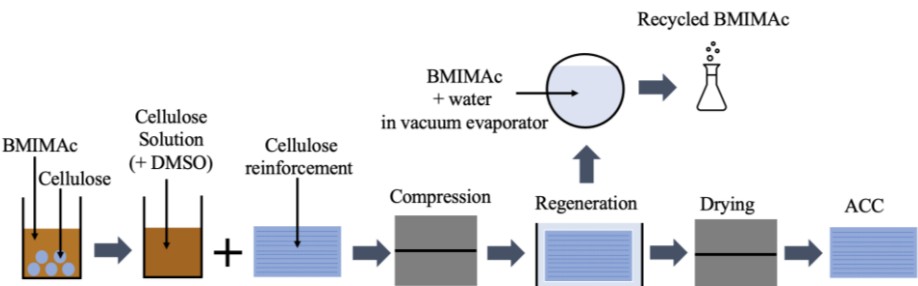

**Figure 4.** Illustration of the two-step method.

The reinforcing materials used for the two-step method with pure [BMIM][Ac] and DMSO were CO and BCO. The cellulose solution contains 53 ± 1 mL DMSO (20 wt% cellulose solution), 226 ± 1 mL [BMIM][Ac] and 3.9 ± 0.1 g cellulose (1.7 wt% cellulose solution). The mixture of [BMIM][Ac], DMSO and cellulose was stirred for 5 min before adding the nonwoven fabric. The dissolution of the cellulose in the fabric was performed at a temperature of 100 °C, 60–90 min or until complete dissolution of the cellulose during stirring. The same pressing procedure from the two-step method with pure [BMIM][Ac] was used (Figure 4). The temperature, pressure and time applied for each sample are presented in Table 2.

All obtained laminates were handled similarly after compression molding. The laminate was placed in a bath with deionized water to regenerate the cellulose by precipitation. A weight was placed on each sample to flatten the formed laminates. The water with the used [BMIM][Ac] was collected to enable the recycling of the [BMIM][Ac]. The laminates were dried in the hydraulic press at 110 °C, with a pressure of 2.5 MPa (100 kN) for 5 to 6 h. A weight was after the pressing placed on the metal plates to reduce the risk of forming an uneven laminate, avoid shrinkage, and enable the cooling of the produced composite.

### 2.2.3. Recycling of [BMIM][Ac]

To enable the recycling of the [BMIM][Ac], a Heidolph rotary evaporator (Heidolph Instruments GmbH & CO, Schwabach, Germany) was used to vacuum distill the IL from the collected water. The temperature was 80 °C, the distillation flask had a constant rotation of 15 rpm and a pressure of 1 bar. The mixture of [BMIM][Ac] and water was continuously refilled. A Rotavapor R-114 and a Waterbath B-480 from Büchi (Flawil, Switzerland) and a Diaphragm vacuum pump MZ 2C from Vacuubrand (Wertheim, Germany) were used to remove the water. The temperature range was between 45 and 80 °C, the pressure was approx. 0.20 bar and the RPM was at a constant speed.

### 2.2.4. Characterization

A Nicolet iS10 from Thermo Scientific (Waltham, MA, USA) with a SMART iTX diamond was used to perform Fourier Transform Infrared Spectroscopy (FTIR) on the pure and collected recycled [BMIM][Ac]. The spectra for the recycled and pure [BMIM][Ac] were compared in order to analyze the peaks due to the O-H groups in the solvent.

The tensile tests were performed in accordance with the ISO 527 standard test method on a Tinius Olsen H10KT (Salfords, UK) machine. The specimens were extended at a rate of 10 mm/min. Each sample had five test specimens tested. Impact tests were performed to detect the impact energy required to fracture a composite. The test was performed

according to ISO 179 using a Zwick test instrument with a 5 J swinging arm. The samples were tested in an edgewise manner and un-notched.

A Nikon SMZ800 stereomicroscope (Tokyo, Japan) was used in order to observe the morphology of the cross-section of a composite and detect voids.

The density of the composites was determined following the hydrostatic weighing method, which is based on Archimedes' principle [15]. Ethanol was used as the liquid medium and the composites' densities were calculated according to Equation (1).

$$\rho_{com} = \frac{W_a}{(W_a - W_e)} \times \rho_e \qquad (1)$$

where $\rho_{com}$: is the density of the composite ($g/cm^3$), $W_a$ is the weight in air (g), $W_e$ is the weight in ethanol (g), and $\rho_e$ is the density of ethanol (0.79 $g/cm^3$).

The measured density is used to be able to calculate the void content of the composite samples. Equation (2) was used for calculating the void content.

$$V = 1 - \frac{\rho_{com}}{\rho_{cot}} \times 100 \qquad (2)$$

where V is the void content (%), $\rho_{com}$ is the density of the composite-measured density ($g/cm^3$), and $\rho_{cot}$ is the density of the cotton-theoretical density (1.54 $g/cm^3$).

### 2.2.5. Statistical Analysis

The statistical analysis of the test results was performed using one-way ANOVA.

## 3. Results and Discussions

### 3.1. Recycling of Ionic Liquid

The removal of the IL by distillation was followed by FTIR analysis during the distillation. The FTIR spectra were observed in the band position 3200–3650 cm$^{-1}$ since the OH groups from the water can be detected in this range. Figure 5 shows the result from the FTIR on the recycled [BMIM][Ac] (RIL). The red line represents pure [BMIM][Ac] (PIL). The purple line (OH_0) in the bottom was taken from the spectra of IL obtained in the first step of distillation. This indicates a large content of OH groups compared to PIL. The green line shows a decrease in OH content after 3 h distillation (OH_3) and the blue line shows a decrease in OH groups after 5 h distillation (OH_5) in the evaporator. Nevertheless, the solution still contained water. The pink line (RIL) shows that OH bands decreased even more (after 7 h) and are closest to PIL. However, the difference between the solvents indicates that small amounts of water still remained in the RIL sample. It might be possible to remove even more OH content by extending the time in the evaporator.

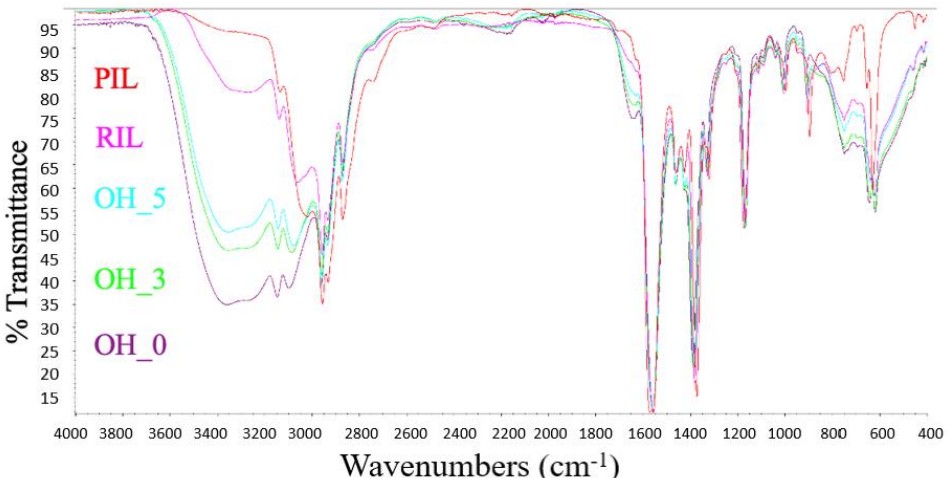

**Figure 5.** Visual decrease of oh-content in the [BMIM][Ac]-water solution.

### 3.2. Mechanical Properties of ACCs

Table 3 shows an overview of the tensile and impact properties of the manufactured ACCs.

**Table 3.** Mechanical properties of manufactured ACCs.

| Sample | Strength (MPa) | E-Modulus (GPa) | Impact Strength (kJ/m$^2$) |
|---|---|---|---|
| CO1 | 42.7 (16.6) | 6.0 (0.7) | 39.8 (5.6) |
| CO2 | 29.5 (16.8) | 2.1 (0.5) | 33.1 (7.2) |
| CO3 | 55.8 (5.7) | 4.1 (1.3) | 22.4 (4.5) |
| BCO1 | 53.5 (5.7) | 4.8 (1.6) | 62.4 (13.1) |
| BCO2 | 39.2 (13.3) | 5.7 (0.2) | 42.0 (8.1) |
| BCO3 | 49.5 (6.9) | 4.5 (0.8) | 34.1 (6.2) |
| RIL CO | 29.1 (6.8) | 4.7 (0.4) | 40.1 (4.5) |

### 3.2.1. Influence of Recycling of Ionic Liquid

When comparing the tensile properties between CO1 and RILCO laminates, both the highest E-modulus (6.0 vs. 4.7 GPa) and tensile strength (42.7 vs. 20.1 MPa) were achieved for CO1. However, the statistical analysis shows no significant difference between the composite specimens ($p$-value > 0.05) which indicates that the choice of pure IL or recycled IL does not have a significant influence on the tensile properties of these samples.

### 3.2.2. Influence of Manufacturing Method

When comparing the ACCs manufactured by the one-step method (CO1) and the two-step method (CO2), the statistical analysis indicates no significant difference in tensile strength ($p$-value > 0.05). The E-modulus, however, had a significant difference between CO1 and CO2 processed composites ($p$-value < 0.05), where CO1 achieved a higher E-modulus. This could be due to the fact that the fiber skin of the cellulose reinforcement is transformed into the matrix phase when using partial dissolution (CO2). It means that the crystal structure of cellulose I is converted to that of cellulose II by the regeneration process. Cellulose I is the highest crystallinity type, containing two co-existing crystal phases: cellulose I$_\alpha$ and cellulose I$_\beta$. The E-modulus of the crystalline regions of the various cellulose polymorphs in the direction of the chain axis is different, which indicates that the polymer skeletons of these polymorphs are entirely different from each other. During the crystallization transition, the skeletal conformations and intramolecular hydrogen bonds are altered [16]. The E-modulus values of cellulose I and II are reported to be 138 and 88, GPa, respectively, [17].

Moreover, statistical analysis indicates no significant difference in impact resistance between composite manufacturing methods, which indicates that there was no influence of the chosen methods on the impact resistance.

### 3.2.3. Influence of Material

The influence of the type of discarded end-of-life denim on tensile and impact properties was studied, where a one-step method with pure [BMIM][Ac] was used with cotton and a blend of cotton. When comparing CO1 and BCO1 laminates, the result was quite similar for the E-modulus. The tensile strength was higher for BCO1 laminates (53.5 vs. 42.7 MPa). The highest E-modulus was obtained by CO1 laminates (6.0 vs. 4.6 GPa). Nonetheless, the significance ($p$-value > 0.05) indicates that there is no influence of the type of denim material when using the one-step method on the tensile strength or the E-modulus. As for the result from comparing pure and recycled [BMIM][Ac], the formed reinforcement used in CO1 and BCO1 has the same structure which can result in the choice of material not influencing the tensile properties. BCO1 achieved higher energy absorption (62.41 kJ/m$^2$) compared to CO1 (39.79 kJ/m$^2$) in the impact test. Statistical analysis confirmed a significant difference

in impact resistance between CO1 and BCO1 laminates (*p*-value < 0.05), which means that the choice of cotton and blend of cotton have an influence on the impact resistance.

### 3.2.4. Influence of Method and Material

ACC specimens manufactured via the two-step method were compared with the specimens manufactured via the two-step method with DMSO as the co-solvent, and also with different denim materials—CO2, CO3, BCO2 and BCO3. BCO2 had the highest E-modulus while CO2 had the lowest. BCO3 showed the highest tensile strength while CO2 continued to have the lowest values. The significance (*p*-value > 0.05) indicates that the tensile strength was not influenced by the combination of method and material, which was the opposite in the statistical analysis of the E-modulus, where the significance level (*p*-value < 0.05) indicates that the combination of material and method could have an influence on the E-modulus. The undissolved blended content in BCO2 and BCO3 could influence the E-modulus since the blended fabric contains, among others, polyester fibers, which can act as reinforcement. The highest energy absorption during impact testing was maintained by BCO2 (42.0 kJ/m$^2$), while CO3 maintained the lowest (22.4 kJ/m$^2$). Statistical analysis verifies that there was no significant difference between denim type materials and process methods (*p*-value > 0.05). The result indicates that the material and method in this case do not influence the impact resistance.

### 3.2.5. Influence of DMSO

The influence of the co-solvent DMSO on the tensile properties and impact resistance was studied for both CO and BCO. This is relevant as DMSO will decrease the viscosity and will therefore have an effect on the impregnation and cellulose dissolving. DMSO is added to decrease the viscosity in the [BMIM][Ac] + CO solution [14]. The statistical ana-lysis was performed to compare between CO2 and CO3, BCO2 and BCO3 and CO3 and BCO3. The reduction of viscosity appears to influence the E-modulus of the produced ACCs. It could be due to improved bonding at the interface [18]. However, the tensile strength was not influenced by the addition of DMSO (*p*-value > 0.05) to CO2-CO3 or BCO2-BCO3. Furthermore, the statistical analysis indicates that there was no influence on the choice of material with DMSO-CO3 and BCO3 for both E-modulus and tensile strength (*p*-value > 0.05). Hence, it is evident that the reinforcement contributes to the tensile properties. In addition, the result from the analysis showed no influence of the DMSO on the impact resistance for either of the comparisons (*p*-value > 0.05).

The results reported by us previously using sheets of denim waste to produce ACCs ([BMIM][Ac])-Denim-Two step method) resulted in a given tensile strength of 26.7 MPa with an E-modulus of 3.5 GPa and an impact resistance of 37 kJ/m$^2$ [19]. It can be concluded that a higher E-modulus and tensile strength were achieved in this research compared to the previously reported research. The difference in the results can depend on the different constructions of reinforcement, as we previously used twill-woven denim and a different matrix phase compared to this research where nonwoven denim fibers were tested. Since the reinforcement influences the mechanical properties, it contributes to the difference in the results.

When comparing our results to the mechanical properties of other ACCs reported in the literature [20], it can be seen that similar results have been achieved (Figure 6). Gindl-Altmutter et al. (2012) produced ACCs by partial dissolution with flax and Lyocell in nonwoven laminates [21], where the flax ACC gave a Young's modulus of 4.6 GPa and a tensile strength of 34 MPa, and the Lyocell ACC gave a Young's modulus of 7.2 GPa and a tensile strength of 78 MPa. The researchers concluded that the ACC produced from flax had lower mechanical properties compared to a conventional composite produced with a flax and epoxy matrix. The ACC produced with Lyocell achieved a similar tensile strength compared to the composite with an epoxy matrix and Lyocell. The results showed that the ionic liquid changes the structure of the cellulosic fibers—in this case the flax fibers—to a degree where it weakens the mechanical properties of the flax ACC [21].

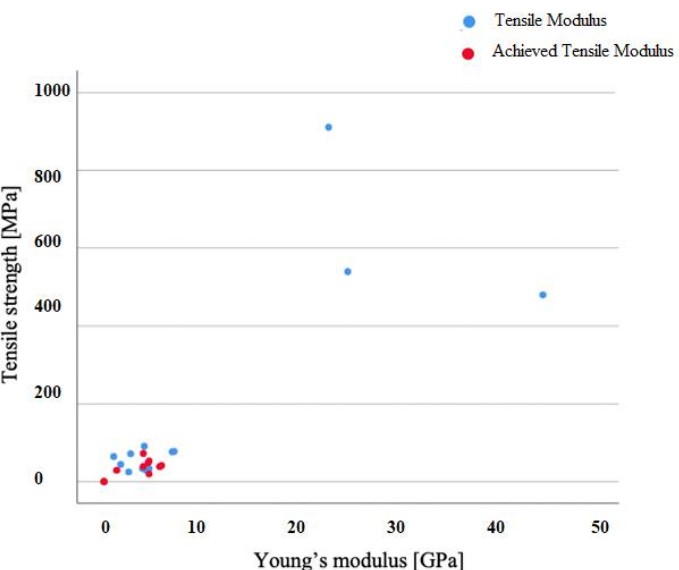

**Figure 6.** Tensile properties of ACCs from literature presented in [20] (blue dots) and measured tensile properties from this study (red dots).

*3.3. Microscopic Analysis*

Figure 7 visualizes the difference between the different denim textile materials and production methods—the one-step method, the two-step method and the two-step method with DMSO. It was possible to imply that the BCO1 laminate had a better-dissolved cross-section compared to the CO1 and BCO2 laminates. However, it was difficult to differentiate between the fibers and layers in the cross-section of the CO3 and BCO3 laminates compared to the other laminates. In the CO1 and BCO1 laminates, it was possible to detect voids, shown in the white circles in Figure 7.

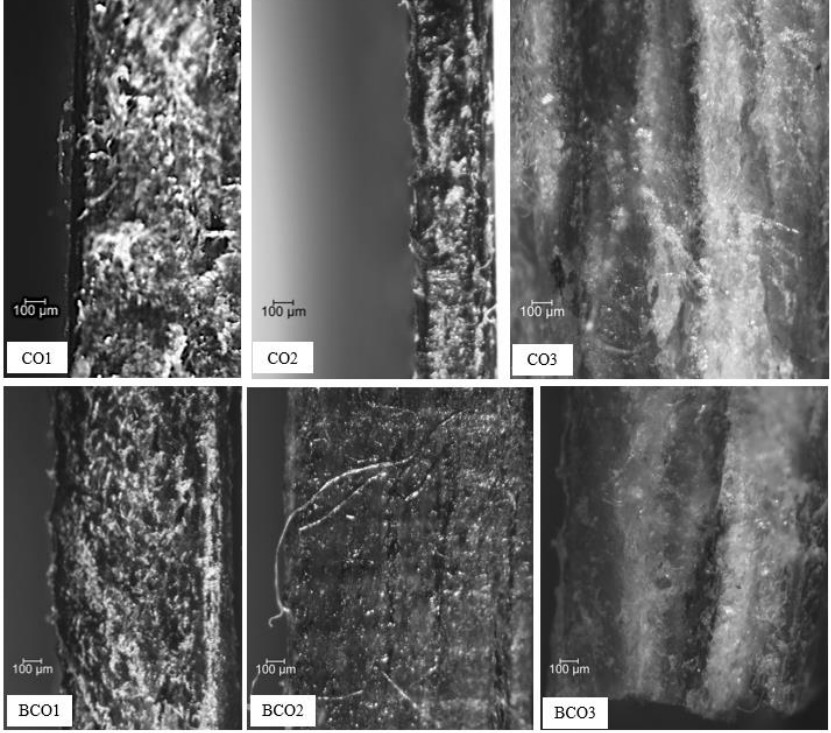

**Figure 7.** Microscopic comparison (×4) between cotton with partial dissolution (CO1); two-step method (CO2); two-step method with Dimethyl Sulfoxide (CO3; and blend cotton with partial dissolution (BCO1); two-step method (BCO2) and two-step method with Dimethyl Sulfoxide (BCO3).

Figure 8 shows the micrographs of cross-sections of laminates produced with partial dissolution—CO1, RILCO and PCO. The cross-section of the CO1 sample shows a uniform surface where the fiber sticks up. In the RILCO sample, it is possible to distinguish layers of nonwoven laminates, as seen in the white circle. Although, it can be seen that some dissolution has occurred.

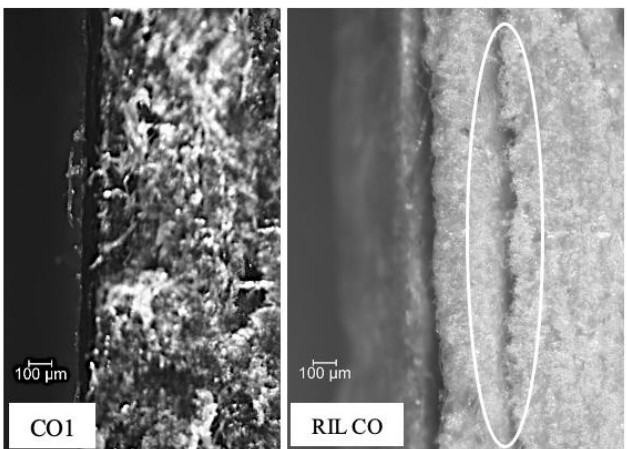

**Figure 8.** Microscopic comparison (×4) between samples done by partial dissolution with pure [BMIM][Ac] and cotton (CO1); recycled [BMIM][Ac] and cotton (RIL CO).

The detected and unobserved voids can depend on the uneven dissolution of the material since the impregnation of [BMIM][Ac] on the nonwoven laminates might be uneven.

### 3.4. Void Content

Table 4 shows the measured density and void content for each composite. The lowest void content was found for the laminates made from 100% cotton denim—CO1 and CO2. The composites made from the blend denim fabric resulted in a slightly higher void content—BCO1, BCO2 and BCO3. Statistical analysis shows that there was a difference between the composite produced with recycled IL and pure IL ($p$-value < 0.05). There was no significant difference ($p$-value > 0.05) for the samples produced with the one-step or two-step methods, pure cotton or blended cotton, which indicates that the method and material, in this case, do not influence the void content.

**Table 4.** Overview of the density of composites and void content.

| Sample | Measured Density [g/cm$^3$] | Void Content [%] |
|--------|-----------------------------|------------------|
| CO1 | 1.43 | 7.45 |
| CO2 | 1.43 | 7.00 |
| CO3 | 1.40 | 8.89 |
| BCO1 | 1.40 | 9.15 |
| BCO2 | 1.40 | 8.79 |
| BCO3 | 1.41 | 8.37 |
| RILCO | 1.36 | 11.46 |

## 4. Conclusions

Today's consumption of textiles generates a large volume of textile waste. Therefore, it is necessary to find solutions to re-use textile waste rather than recycling fibers into new fibers. Research using pre- and post-consumer textiles in composites is ongoing in an interesting direction. The main objective of the present study was the assessment of the properties of ACCs from discarded denim produced with one-step and two-step

methods (with and without DMSO). Discarded denim fabrics with 100% cotton and blend materials were converted to nonwoven fabrics by shredding, carding and needle punching. The produced nonwoven laminates were used as the reinforcement in the composites while dissolved cellulose in an ionic liquid (IL), 1-butyl-3-methyl imidazolium acetate ([BMIM][Ac]), was used as the matrix phase. The matrix was then regenerated by removal of the IL by washing to form the composite. The washed-out IL was collected and recycled in order to study the effect of its reuse as a recycled cellulose solvent on the mechanical properties of ACCs. The results showed that the tensile strength of the composites was not highly affected by ACCs manufacturing methods. However, there was a significant difference between the E-modulus of ACCs produced with the one-step method (6 GPa) and the two-step method (2.1 GPa). Moreover, the E-modulus of the ACCs was influenced by the addition of DMSO as a co-solvent for IL while the tensile strength was not affected. The choice of pure IL and recycled IL does not have a significant influence on the tensile strength, E-modulus or impact resistance of the produced ACCs. From the impact test results, it was evident that there was not a significant difference between the impact resistance of ACCs manufactured with the one-step ($39.79 \pm 6.85$ kJ/m$^2$) and two-step ($33.07 \pm 8.82$ kJ/m$^2$) methods.

It can be concluded that the achieved values for the mechanical properties can be compared with other ACCs reported in the literature. The results from the research imply that it is possible to find a new purpose for recycled textiles in the form of composites. With this approach, it is possible to avoid the unnecessary disposal of textiles containing cellulose.

**Author Contributions:** Conceptualization, M.S. and B.B.; methodology, M.S., B.B., B.J. and N.K.; validation, M.S., B.B. and B.J.; formal analysis, B.B. and B.J.; investigation, B.B. and B.J.; resources, M.S. and B.B.; data curation, M.S., B.B., B.J. and N.K.; writing—original draft preparation, B.B.; writing—review and editing, M.S. and B.B.; visualization, B.B. and B.J.; supervision, M.S. and B.B.; project administration, M.S. and B.B.; funding acquisition, M.S. All authors have read and agreed to the published version of the manuscript.

**Funding:** This research was funded by Formas grant number 2016-00920.

**Institutional Review Board Statement:** Not applicable.

**Informed Consent Statement:** Not applicable.

**Acknowledgments:** The authors acknowledge the financial support from FORMAS—A Swedish Research Council for Sustainable Development (Grant 2016-00920).

**Conflicts of Interest:** The authors declare no conflict of interest.

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
