# Peer review of "All-Cellulose Composites Properties from Pre- and Post-Consumer Denim Wastes: Comparative Study"

_jcs, doi:10.3390/jcs6050130_

Round 1

Reviewer 1 Report

Authors take up an important topic related to textile recycling. After reading the article, the following comments come to my mind:

  1. Maybe the title "All-Cellulose Composites Properties Comparative Study" would be better?
  2. The abstract is badly written. It is not typical abstract. This is meant to be a summary of the research methodology. The introductory background information that leads into a statement of the article aim is missing. The methodology is described unclearly and too broadly (lines 11-21). The most important findings are missing. There is no general conclusion in the abstract.
  3. Line 14: instead of "all-cellulose composites" it should be "ACCs"
  4. Line 52-54: The two compared methods of creating ACCs are very vaguely named. I suggest: complete dissolution method (two step method) and partial dissolution method (one-step method).
  5. Lines 63-65: target declared: "The aim of this study is to create ACCs produced from discarded textiles, in the form of nonwoven from denim waste together with ionic liquid to investigate if the method can be used to recycle textiles into composites for structural applications" The article shows something else. The article compares ACCs produced by two methods and with different parameters. Additionally the aim: "to create ACCs produced from discarded textiles" is little scientific. What does "structural applications" mean. Please detail this.
  6. Line 54: reference error. Line 113: reference error. Line 129: reference error. Line 141: reference error Line 146: reference error. Line 159: reference error Line 313: reference error. Line 329: reference error Line 341: reference error
  7. Line 362: The "Conclusions" section needs to be completed. It stated: "It could be concluded that the achieved values for the mechanical properties can be compared with other ACCs reported in literature." This is not a research conclusion. Everything can be compared with each other. But from this comparison something interesting should arise. The summarized key research findings are missing. The broader implications of the findings are also missing. This section also says "The results from the research implies that it is possible to find a new purpose for recycled textiles in the form of composites." This is too general. Please indicate any such new potential use for ACC. Or maybe you can indicate future research directions? That would justify  the importance of the study.

Sincerelly

Author Response

Response to reviewers

Reviewer #1:

  1. Maybe the title "All-Cellulose Composites Properties Comparative Study" would be better?
    • The title was changed to “All-cellulose composites properties from pre- and post-consumer denim wastes: Comparative Study“.
  2. The abstract is badly written. It is not typical abstract. This is meant to be a summary of the research methodology. The introductory background information that leads into a statement of the article aim is missing. The methodology is described unclearly and too broadly (lines 11-21). The most important findings are missing. There is no general conclusion in the abstract.
    • The whole manuscript text is revised. The abstract and conclusion parts were revised as well.
  3. Line 14: instead of "all-cellulose composites" it should be "ACCs"
    • It was changed in the text.
  4. Line 52-54: The two compared methods of creating ACCs are very vaguely named. I suggest: complete dissolution method (two step method) and partial dissolution method (one-step method).
    • The text was revised.
  5. Lines 63-65: target declared: "The aim of this study is to create ACCs produced from discarded textiles, in the form of nonwoven from denim waste together with ionic liquid to investigate if the method can be used to recycle textiles into composites for structural applications" The article shows something else. The article compares ACCs produced by two methods and with different parameters. Additionally the aim: "to create ACCs produced from discarded textiles" is little scientific. What does "structural applications" mean. Please detail this.
    • Structural application was removed from the text and the aim was revised.
  6. Line 54: reference error. Line 113: reference error. Line 129: reference error. Line 141: reference error Line 146: reference error. Line 159: reference error Line 313: reference error. Line 329: reference error Line 341: reference error
    • The text was edited.
  7. Line 362: The "Conclusions" section needs to be completed. It stated: "It could be concluded that the achieved values for the mechanical properties can be compared with other ACCs reported in literature." This is not a research conclusion. Everything can be compared with each other. But from this comparison something interesting should arise. The summarized key research findings are missing. The broader implications of the findings are also missing. This section also says "The results from the research implies that it is possible to find a new purpose for recycled textiles in the form of composites." This is too general. Please indicate any such new potential use for ACC. Or maybe you can indicate future research directions? That would justify  the importance of the study.
    • The conclusion was revised.

Reviewer 2 Report

The authors provide an interesting aspect of all-cellulose composite (ACC) research, namely the ability to recycle and reuse cellulose-based fashion textiles. While the idea is interesting the execution and presentation need more work to be interesting to the readers of this journal. 

The introduction is well written but would benefit from providing more information on the chemical composition of denim as well as the mechanical properties of the fibre, so the reader can better contextualise the presented results. 

It is very commendable that the authors provide p-values for their statistical test, but this would be more useful if the used statistical test would also be provided. More importantly, though, in the methods, the authors describe variations of tested materials (CO, BCO, RILCO), method (Partial dissolution, two-step), Solvent and solution, fibre content, pressure and processing time with at least 2 levels. A full factorial analysis of those factors would require at least 64 different samples to come to any statistically relevant conclusions. However, the authors test only 7 different samples. If the authors chose the used settings based on Taguchi or a similar approach to rescue the number of necessary experiments, it is not mentioned in the manuscript. Yet, the bulk of the discussion addresses differences in these factors. It is highly questionable if the made observations and conclusions can be supported by the available data. More experiments will need to be conducted to comprehensively discuss all of these differences. Alternatively, the authors could explore more robust statistical methods for their analysis or reduce the number of discussed aspects. This is resembled in the results and discussion. The authors make hardly any attempt to discuss their results, very few of them are put in the context of other published literature, and in at least one aspect this is done incorrectly. The whole results and discussion section should be reworked to be precise, concise and to put the generated results into context.

Additionally, most of the figures are of poor quality, none of the cross-references to figures and tables that must have been in the word document work, which is irritating when reading, and some language errors need addressing. Most are pointed out below, but I would recommend the manuscript gets proofread by a native speaker.

More detailed comments follow below:

L69:

The materials section is lacking the necessary detailed information on the fraction of polyester and elastane in the blended material. Later on, the impact properties from a blend based composite are highlighted but it remains fully unclear how much the elastane and/polyester might be contributing. There is also no information available on the properties of the denim fibres. Without this information, it is impossible to judge whether the chosen processing routes provide an optimal solution to achieve high mechanical properties. 

L77:

The authors claim the co-solvent reduces the viscosity of the ionic liquid. While this is true, it appears the co-solvent was only added to cellulose solutions (Table 2) not the IL itself. Therefore, the authors should provide viscosity data for this solution to better understand how much of a reduction was achieved and how similar/different the solution was in viscosity to the IL to better judge any differences in textile impregnation.

L 78:

The authors state that adding co-solvent increases the difficulty of recycling. However, in the remainder of the manuscript, this point is not addressed. Why is it more difficult? If so, is there data to support this claim? 

L93:

The authors report an average fibre length and speculative explanation as why the carding process affects this average length. It would be more helpful to provide a size distribution or at least to provide not only the average but also the median and standard deviation to better judge the change in fibre length. 

L103:

The authors claim one cycle of carding produces "acceptable" webs. Please provide a definition of acceptable. Which criteria were applied to judge the quality of the web and how were those evaluated? 

L114:

The authors claim the fibre content of ACCs produced via partial dissolution cannot be calculated. However, this has been demonstrated for example in:

Dormanns, Jan W., et al. "Solvent infusion processing of all-cellulose composite laminates using an aqueous NaOH/urea solvent system." Composites Part A: Applied Science and Manufacturing 82 (2016): 130-140.

Additionally, there are many publications online that show SEM images of ACCs through which it is typically possible to distinguish between fibre and matrix phase, so SEM would be the recommendable technique to confirm observations. It might also be possible to comment on the amount of created matrix by measuring the crystallinity of the made ACCs as dissolved and regenerated cellulose typically is of lower crystallinity. 

L135:

The authors claim nonwovens were impregnated by pouring the ionic liquid onto them. Was the IL not worked into the textile in any way using hands/rollers/etc? It is hard to imagine good penetration and distribution of the textile was achieved by simply pouring the IL. How was impregnation confirmed?

L144:

The authors claim complete dissolution occurred after 60-90 minutes. How was this confirmed?

L154:

 “The mixture of [BMIM][Ac], DMSO and cellulose were stirred for 5 min before adding the fabric.” Was the fabric added to the mixture or was the mixture added to the fabric?

L160f:

How long were the laminates kept in the water bath? Was the water changed? How did the authors check if all of the IL was removed from the laminate? It is well known that full removal of an IL is challenging (Gericke, Martin, Pedro Fardim, and Thomas Heinze. "Ionic liquids—Promising but challenging solvents for homogeneous derivatization of cellulose." Molecules 17.6 (2012): 7458-7502.)  so the authors should include a statement on their own process.

L166:

How does placing a weight on a sample avoid shrinkage and enable cooling?

L174:

The authors claim “the last content of water” was removed but in L216 they state “it might be possible to remove even more OH content by extending the time in the evaporator.” Those two statements directly contradict each other. Please clarify.

L188:

How were the samples prepared for microscopy? Presumably, the cross-section was cut, but without specifying how, it is not possible rule out artefacts from the sample preparation. The poor quality of the images in Figure 7 suggests inadequate sample preparation.

Discussion point 3.1:

 Figure 5 is of poor quality. The figure is too low resolution and some of the text is too small to read easily. The authors only discuss the region from 3200-3650 cm-1, I, therefore, recommend providing a subsection of the covered spectrum to better see the described differences.

FTIR is not ideal for confirming recyclability of an IL; by-products and impurities might be generated during cellulose dissolution that are better more reliably detected using NMR (Gericke, Martin, Pedro Fardim, and Thomas Heinze. "Ionic liquids—Promising but challenging solvents for homogeneous derivatization of cellulose." Molecules 17.6 (2012): 7458-7502., Elsayed, Sherif, et al. "Recycling of superbase-based ionic liquid solvents for the production of textile-grade regenerated cellulose fibers in the lyocell process." ACS Sustainable Chemistry & Engineering 8.37 (2020): 14217-14227.) so the authors should comment if their analysis can reliably state that no impurities were generated during their process which would be a requirement for claiming recyclability.

Discussion point 3.2/Table 4:

The authors should provide an explanation for the very high standard deviations (>50% for sample CO2, >25% for BCO1 and BCO2). What is causing those variations? Could they be an indicator of poorly processed samples?

 E-modulus should be Young’s modulus throughout the whole manuscript.

No discussion of this section is provided. It is solely a presentation of results.

Discussion point 3.3:

“When comparing the tensile properties between CO1 and RILCO laminates, both highest E-modulus (6.0 vs 4.7 GPa) and tensile strength (42.7 vs 20.1 MPa) was achieved for the composite made from the 100% cotton denim fabric processed by the partial dissolution process (CO1).”

This sentence needs to be phrased differently. How does the RILCO sample have the highest Young’s modulus and compared to what? Based on which values do the authors conclude “that the choice of pure [BMIM][Ac] or recycled [BMIM][Ac] does not have a significant influence on the tensile properties of these samples.” when the values are different to each other. Which values are being compared here?

Discussion point 3.4:

The authors speculate that a higher Young’s modulus is the result of better interfacial adhesion. However, interfacial adhesion typically causes increased tensile strength as it allows a better load transfer from weaker matrix to stronger fibre. It, therefore, is an insufficient explanation for the higher Young’s modulus of sample CO1.

“Moreover, statistical analysis indicates no significant difference in impact resistance between manufacturing methods, which indicates that there was no influence of the chosen methods on the impact resistance”. See the point raised above, due to many different factors this conclusion cannot be drawn here as the authors do not only compare manufacturing methods.

Discussion point 3.5:

As above, the number of samples and their differences do not allow to draw these conclusions.

Discussion point 3.6:

How is “Influence of method and material” different from sections 3.4 “Influence of manufacturing method” and section 3.5 “Influence of material”? The authors should really consider merging some if not all of these sections as they are very brief, the discussions are far from comprehensive and the lack of tested samples does not allow for those distinctions anyway.

L307:

Wrong use of the word “mean”

L309:

 “It could be interesting to try the nonwoven laminated from discarded textiles with a bio-based resin to compare the mechanical properties.” Why did the authors not do that then? If it is beyond the scope of this work, why mention it?

Figure 6:

A poor quality figure, symbols are too small, the figure is too pixelated and the legend is non-sensical. It is also not clear from which references the comparison points were generated.

L318:

The manuscript does not provide any data for flexural strength or stiffness so the comparison to the reference is void. Flexural properties are measured via a bending test, not through tensile testing.

Discussion Point 3.8:

The provided figure is of too poor quality to see any differences in the samples. However, this is more likely the result of poor sample preparation rather than the effect of processing. I strongly recommend using an SEM for some/all of this analysis. At the given magnification it is not possible to see individual fibres, so one cannot judge how well they are covered by the matrix phase and where microvoids might be.

Discussion 3.9:

No explanation is offered for the overall high, and more importantly, different void content of the samples. This is not a discussion.

Author Response

Reviewer #2:

The introduction is well written but would benefit from providing more information on the chemical composition of denim as well as the mechanical properties of the fibre, so the reader can better contextualise the presented results. 

  • More information was included in the text.
  • Since the fabrics used for today’s denim jeans vary (100% cotton, 60%cotton/40% polyester, 50% cotton/50% polyester, 60% polyester/40% cotton, blend of cotton, nylon and polyester), it is difficult to provide information regarding mechanical properties.

It is very commendable that the authors provide p-values for their statistical test, but this would be more useful if the used statistical test would also be provided. More importantly, though, in the methods, the authors describe variations of tested materials (CO, BCO, RILCO), method (Partial dissolution, two-step), Solvent and solution, fibre content, pressure and processing time with at least 2 levels. A full factorial analysis of those factors would require at least 64 different samples to come to any statistically relevant conclusions. However, the authors test only 7 different samples. If the authors chose the used settings based on Taguchi or a similar approach to rescue the number of necessary experiments, it is not mentioned in the manuscript. Yet, the bulk of the discussion addresses differences in these factors. It is highly questionable if the made observations and conclusions can be supported by the available data. More experiments will need to be conducted to comprehensively discuss all of these differences. Alternatively, the authors could explore more robust statistical methods for their analysis or reduce the number of discussed aspects. This is resembled in the results and discussion. The authors make hardly any attempt to discuss their results, very few of them are put in the context of other published literature, and in at least one aspect this is done incorrectly. The whole results and discussion section should be reworked to be precise, concise and to put the generated results into context.

Additionally, most of the figures are of poor quality, none of the cross-references to figures and tables that must have been in the word document work, which is irritating when reading, and some language errors need addressing. Most are pointed out below, but I would recommend the manuscript gets proofread by a native speaker.

More detailed comments follow below:

L69: The materials section is lacking the necessary detailed information on the fraction of polyester and elastane in the blended material. Later on, the impact properties from a blend based composite are highlighted but it remains fully unclear how much the elastane and/polyester might be contributing. There is also no information available on the properties of the denim fibres. Without this information, it is impossible to judge whether the chosen processing routes provide an optimal solution to achieve high mechanical properties. 

  • It is preliminary study to investigate if the blended material including polyester and elastane has effect on ACCs or not even the amount of them is very low. In future work, your comments will be implemented, and more study will be done in details in terms of contribution of blended materials in ACCs mechanical properties.

L77: The authors claim the co-solvent reduces the viscosity of the ionic liquid. While this is true, it appears the co-solvent was only added to cellulose solutions (Table 2) not the IL itself. Therefore, the authors should provide viscosity data for this solution to better understand how much of a reduction was achieved and how similar/different the solution was in viscosity to the IL to better judge any differences in textile impregnation.

  • Good point, the viscosity of the cellulose solution including IL and DMSO will be measured in the future studies.  

L 78: The authors state that adding co-solvent increases the difficulty of recycling. However, in the remainder of the manuscript, this point is not addressed. Why is it more difficult? If so, is there data to support this claim? 

  • The difficulty of recycling is related to the fact that DMSO is not environmentally friendly material compared to IL. Therefore, combination of IL and DMSO would make some challenges regarding the recycling of IL. In this study, we did not investigate the possibilities/methods of separation of DMSO and IL.

L93: The authors report an average fibre length and speculative explanation as why the carding process affects this average length. It would be more helpful to provide a size distribution or at least to provide not only the average but also the median and standard deviation to better judge the change in fibre length. 

  • Unfortunately, this information is not available, but it will be considered in the future studies.

L103: The authors claim one cycle of carding produces "acceptable" webs. Please provide a definition of acceptable. Which criteria were applied to judge the quality of the web and how were those evaluated? 

  • This part was revised as: The carding was done only once due to difficulties during carding where the short fibers got stuck on the fiber drum during carding.

L114: The authors claim the fibre content of ACCs produced via partial dissolution cannot be calculated. However, this has been demonstrated for example in:

Dormanns, Jan W., et al. "Solvent infusion processing of all-cellulose composite laminates using an aqueous NaOH/urea solvent system." Composites Part A: Applied Science and Manufacturing 82 (2016): 130-140.

Additionally, there are many publications online that show SEM images of ACCs through which it is typically possible to distinguish between fibre and matrix phase, so SEM would be the recommendable technique to confirm observations. It might also be possible to comment on the amount of created matrix by measuring the crystallinity of the made ACCs as dissolved and regenerated cellulose typically is of lower crystallinity. 

  • Thanks for your comment. It will be considered in the future studies.

L135: The authors claim nonwovens were impregnated by pouring the ionic liquid onto them. Was the IL not worked into the textile in any way using hands/rollers/etc? It is hard to imagine good penetration and distribution of the textile was achieved by simply pouring the IL. How was impregnation confirmed?

  • The viscosity of IL was good enough to impregnate the reinforcement. The process continued afterwards with pressing the laminates with different temperature, pressure and time as presented in Table 2.

L144: The authors claim complete dissolution occurred after 60-90 minutes. How was this confirmed?

  • The cellulose solution was homogenous.

L154:  “The mixture of [BMIM][Ac], DMSO and cellulose were stirred for 5 min before adding the fabric.” Was the fabric added to the mixture or was the mixture added to the fabric?

  • The mixture added to the fabric.

L160: How long were the laminates kept in the water bath? Was the water changed? How did the authors check if all of the IL was removed from the laminate? It is well known that full removal of an IL is challenging (Gericke, Martin, Pedro Fardim, and Thomas Heinze. "Ionic liquids—Promising but challenging solvents for homogeneous derivatization of cellulose." Molecules 17.6 (2012): 7458-7502.)  so the authors should include a statement on their own process.

  • All samples were handled equivalent after pressing. The sample was placed in a box with deionized water for regeneration. A weight was placed on each sample. The water was changed frequently each day, until the [BMIM][Ac] content was removed from the composites.

L166: How does placing a weight on a sample avoid shrinkage and enable cooling?

L174: The authors claim “the last content of water” was removed but in L216 they state “it might be possible to remove even more OH content by extending the time in the evaporator.” Those two statements directly contradict each other. Please clarify.

  • The text was revised.

L188: How were the samples prepared for microscopy? Presumably, the cross-section was cut, but without specifying how, it is not possible rule out artefacts from the sample preparation. The poor quality of the images in Figure 7 suggests inadequate sample preparation.

  • The cross section was cut.

Discussion point 3.1:

 Figure 5 is of poor quality. The figure is too low resolution and some of the text is too small to read easily. The authors only discuss the region from 3200-3650 cm-1, I, therefore, recommend providing a subsection of the covered spectrum to better see the described differences.

FTIR is not ideal for confirming recyclability of an IL; by-products and impurities might be generated during cellulose dissolution that are better more reliably detected using NMR (Gericke, Martin, Pedro Fardim, and Thomas Heinze. "Ionic liquids—Promising but challenging solvents for homogeneous derivatization of cellulose." Molecules 17.6 (2012): 7458-7502., Elsayed, Sherif, et al. "Recycling of superbase-based ionic liquid solvents for the production of textile-grade regenerated cellulose fibers in the lyocell process." ACS Sustainable Chemistry & Engineering 8.37 (2020): 14217-14227.) so the authors should comment if their analysis can reliably state that no impurities were generated during their process which would be a requirement for claiming recyclability.

  • The figure was substituted by new one. We will use NMR in the further studies, to determine the recyclability of the Il.

Discussion point 3.2/Table 4:

The authors should provide an explanation for the very high standard deviations (>50% for sample CO2, >25% for BCO1 and BCO2). What is causing those variations? Could they be an indicator of poorly processed samples?

  • It might be due to poor impregnation process, porosity, etc.

 E-modulus should be Young’s modulus throughout the whole manuscript.

No discussion of this section is provided. It is solely a presentation of results.

Discussion point 3.3:

“When comparing the tensile properties between CO1 and RILCO laminates, both highest E-modulus (6.0 vs 4.7 GPa) and tensile strength (42.7 vs 20.1 MPa) was achieved for the composite made from the 100% cotton denim fabric processed by the partial dissolution process (CO1).”

This sentence needs to be phrased differently. How does the RILCO sample have the highest Young’s modulus and compared to what? Based on which values do the authors conclude “that the choice of pure [BMIM][Ac] or recycled [BMIM][Ac] does not have a significant influence on the tensile properties of these samples.” when the values are different to each other. Which values are being compared here?

  • Tensile properties (strength and modules) were analyzed statistically as mentioned in the beginning of the paragraph.
  • The text was revised as bellow:

“When comparing the tensile properties between CO1 and RILCO laminates, both highest E-modulus (6.0 vs 4.7 GPa) and tensile strength (42.7 vs 20.1 MPa) were achieved for CO1. However, the statistical analysis shows no significant difference between the composite specimens (p-value > 0.05) which indicates that the choice of pure IL or recycled Il does not have a significant influence on the tensile properties of these samples”

Discussion point 3.4:

The authors speculate that a higher Young’s modulus is the result of better interfacial adhesion. However, interfacial adhesion typically causes increased tensile strength as it allows a better load transfer from weaker matrix to stronger fibre. It, therefore, is an insufficient explanation for the higher Young’s modulus of sample CO1.

  • The text was revised as bellow:

When comparing the ACCs manufactured by one step method (CO1) and two step method (CO2), the statistical analysis indicates no significant difference in tensile strength (p-value > 0.05). The E-modulus, however had a significant difference between CO1 and CO2 processed composites (p-value < 0.05), where CO1 achieved a higher E-modulus. This could be due to the fact that the fiber skin of the cellulose reinforcement is transformed into matrix phase when using partial dissolution (CO2). It means that the crystal structure of cellulose I is converted to that of cellulose II by regeration process. cellulose I, is the highest crystallinity type, containing two co-existing crystal phases, cellulose Iα and cellulose Iβ. The elastic modulus of the crystalline regions of the various cellulose polymorphs in the direction of the chain axis are different which indicates that the polymer skeletons of these polymorphs are entirely different from each other. During the crystallisation transition, the skeletal conformations and intramolecular hydrogen bonds are altered [16]. The E-modulus values of cellulose I and II are reported to be 138 and 88, GPa respectively [17].

“Moreover, statistical analysis indicates no significant difference in impact resistance between manufacturing methods, which indicates that there was no influence of the chosen methods on the impact resistance”. See the point raised above, due to many different factors this conclusion cannot be drawn here as the authors do not only compare manufacturing methods.

  • CO1 was prepared with on-step method and CO2 with two-step method. These two samples were compared in this part.

Discussion point 3.5:

As above, the number of samples and their differences do not allow to draw these conclusions.

  • The experimental set up was not conducted used a comprehensive factorial design, which would have been very good of course. So therefore, the conclusions can be speculative, but they seem to clearly be indicative.

Discussion point 3.6:

How is “Influence of method and material” different from sections 3.4 “Influence of manufacturing method” and section 3.5 “Influence of material”? The authors should really consider merging some if not all of these sections as they are very brief, the discussions are far from comprehensive and the lack of tested samples does not allow for those distinctions anyway.

  • Since several parameters were evaluated in this study, the influence of them were discussed separately.

L307: Wrong use of the word “mean”

  • The text was revised to “The resultes showed …”.

L309: “It could be interesting to try the nonwoven laminated from discarded textiles with a bio-based resin to compare the mechanical properties.” Why did the authors not do that then? If it is beyond the scope of this work, why mention it?

  • It is removed from the text since it was not relevant in this part.

Figure 6:

A poor quality figure, symbols are too small, the figure is too pixelated and the legend is non-sensical. It is also not clear from which references the comparison points were generated.

  • Figure was replaced with higher quality picture.
  • Referenced 20 was added to the test:

Fig 6: Tensile properties of ACCs from literature presented in [21] (blue dots) and measured tensile properties from this study (red dots).

L318: The manuscript does not provide any data for flexural strength or stiffness so the comparison to the reference is void. Flexural properties are measured via a bending test, not through tensile testing.

  • The flexural test results were not included in this manuscript therefore the text was revised.

Discussion Point 3.8:

The provided figure is of too poor quality to see any differences in the samples. However, this is more likely the result of poor sample preparation rather than the effect of processing. I strongly recommend using an SEM for some/all of this analysis. At the given magnification it is not possible to see individual fibres, so one cannot judge how well they are covered by the matrix phase and where microvoids might be.

  • Unfortunately, new SEM pictures cannot be provided in this manuscript, but this will be done in the further studies.

Discussion 3.9:

No explanation is offered for the overall high, and more importantly, different void content of the samples. This is not a discussion.

  • The aim of this test was to evaluate if the applied methods and materials including cotton, cotton blending, recycling of IL have effect on porosity or not. The results were discussed in this part.

Round 2

Reviewer 1 Report

I am pleased to note that the manuscript has been significantly improved, the Abstract is more specific, the purpose of the article has been better articulated, and the conclusions are also improved. Minor bugs and semantic inconsistencies have been fixed. Two comments:

  1. Article title: Perhaps it would be better "study" in the plural form. Both research and various analyzes were performed, not a single "study" ("All-cellulose composites properties from pre- and post-consumer denim wastes: Comparative Studies")
  2. In line 377 Authors wrote, "It could be concluded that the achieved values for the mechanical properties can be compared with other ACCs reported in literature". I suggest writing, "The achieved values for the mechanical properties are similar to other ACCs reported in the literature".

In my opinion, the article is suitable for publication

Sincerelly,